

# Analyzing bacterial community in pit mud of Yibin Baijiu in China using high throughput sequencing

Li Chen[1,*], Yuzhu Li[2,*], Lei Jin[3], Li He[2], Xiaolin Ao[2], Shuliang Liu[2], Yong Yang[2], Aiping Liu[2], Shujuan Chen[2] and Likou Zou[3]

[1] Yibin Products Quality Supervision and Inspection Institute, Yibin, Sichuan, China
[2] College of Food Science, Sichuan Agricultural University, Ya'an, Sichuan, China
[3] College of Resources, Sichuan Agricultural University, Chengdu, Sichuan, China
[*] These authors contributed equally to this work.

## ABSTRACT

"Yibin Baijiu" (YB) is a special Chinese strong-aroma Baijiu (CSAB) that originated in Yibin, a city in western China. YB is fermented in cellars lined with pit mud (PM), the microbiota in which may affect YB quality. In this study, high throughput sequencing of the 16S rRNA gene was used to demonstrate the bacterial community structure and diversity in PM of YB. In addition, the physicochemical characteristics of PM were also analyzed, including moisture content, pH, and available phosphorous, ammonia nitrogen, and humic acid levels. Results showed that Firmicutes was the dominant phylum in all PM samples with abundance > 70.0%, followed by Euryarchaeota (11.3%), Bacteroidetes (6.5%), Synergistetes (3.0%), Actinobacteria (1.4%), and Proteobacteria (1.2%). Furthermore, 14 different genera with average relative abundance of > 1% were detected. The Chao1 and Shannon indexes did not vary significantly between the sub-layer and middle-layer PM ($P > 0.05$). However, Linear discriminant analysis Effect Size (LEfSe) analysis showed that the relative abundance of *Lactobacillus* in the sub-layer PM was significantly higher than in middle-layer PM. pH differed significantly ($P < 0.05$) between the two groups. Canonical correspondence analysis revealed that bacterial community in PM correlated significantly with available phosphorous content and pH. Our study provides basic data for further elucidating the diversity of microbiota in the PM of YB and the potential mechanism of Baijiu production.

## INTRODUCTION

Baijiu is one of the six most famous distilled liquors worldwide (*Fan & Qian, 2006*; *Jin, Zhu & Xu, 2017*; *McGovern et al., 2004*). It includes four main aroma types: strong, light, sauce, and rice. Among these, the Chinese strong-aroma Baijiu (CSAB) accounts for 70% of the total Baijiu production (*Liu & Sun, 2018*; *Liu et al., 2017b*) . Yibin is one of the birth places of CSAB culture (*Zou, Zhao & Luo, 2018*; *You et al., 2016*), and it is located at the confluence of the Min River, Jinsha River, and Yangtze River, which has good water quality. In addition, it has unique weak acid clay soil with strong viscosity, good water retention

Corresponding authors
Shujuan Chen,
chenshujuan1@163.com
Likou Zou, zoulikou@sicau.edu.cn

property, and abundant minerals, which makes it an excellent natural material for making cellars (*Tang et al., 2012*). Furthermore, Yibin enjoys a mid-subtropical humid monsoon climate, which is perfect for the growth of liquor-making microbes (*Zhao et al., 2013*). Owing to these advantages, Yibin is well-suited for the development of uniquely flavored "Yibin Baijiu" (YB). It is noteworthy that YB has been successfully declared the national geographical indication product of 2010 (*Yang et al., 2017*).

YB is produced by distilling mixed and fermented grains such as sorghum, glutinous rice, rice, wheat, and corn, and is characterized by sweetness, strong aroma, and lasting aftertaste (*Zheng et al., 2013*; *Zheng & Han, 2016*). YB is produced via a traditional solid-state fermentation process that occurs in fermentation pits (approximate dimensions: length, 2.0–3.5 m; width, 2.0–3.0 m; depth, 2.3–2.5 m) (*Barrios-González, 2012*), the inner walls of which are covered with a thick layer of pit mud (PM). The PM is a complex ecosystem consisting of a variety of microbial communities, such as bacteria, archaea, and fungi (*Liu et al., 2017c*). Bacterial communities in PM can break down macromolecules into small peptides and monosaccharides, which produce aromatic compounds responsible for the product flavor (*Tao et al., 2014a*; *Zhao et al., 2012*). However, the underlying mechanism remains to be identified. To understand the role of the microorganisms in YB fermentation and optimize the brewing technology, the composition and diversity of the microbial population in PM has to be investigated.

Since the 1960s, researchers have used traditional culture methods for identifying the microbiota in PM (*Wu et al., 1980*). *Clostridium, Bacillus, Pseudomonas,* and *Sporolactobacillus* were isolated and identified using traditional microbial classification and identification methods (*Yue et al., 2007*). However, these methods detected <1% of culturable microorganisms, which failed to reveal the complete structural characteristics of the PM microbial ecosystem (*Amann, Ludwig & Schleifer, 1995*). Subsequently, molecular methods, such as polymerase chain reaction-denaturing gradient gel electrophoresis (PCR-DGGE) (*Deng et al., 2012*; *Huang et al., 2017*), phospholipid fatty acid (PLFA) analysis (*Zheng et al., 2013*), and clone library analysis of the 16S rRNA gene (*Ye et al., 2013*), have been used extensively. These methods not only circumvented the tedious process of microbial cultivation, but also contributed to our understanding of the PM ecosystem. Thus, Firmicutes, Proteobacteria, Bacteroidetes, Actinobacteria, and Synergistetes were found to be predominant in many PM bacterial communities (*Ding et al., 2014a*; *Liang et al., 2016*). Although these molecular methods provided the first description of the diverse microbial populations in PM, the extent of bacterial diversity remains unexplored. Currently, 16S rRNA gene high throughput sequencing is being used for microbial identification (*Liu et al., 2017a*; *Zheng et al., 2015*). However, studies investigating the bacterial community structure and diversity in PM of YB using high-throughput sequencing strategies are lacking. Furthermore, Yibin has a brewing history of nearly a thousand years, suitable ecological factors, and relatively special brewing techniques, which have gradually contributed to the formation of a unique and stable bacterial community structure in PM. Therefore, considering the complexity of PM microorganisms, it is necessary to study the PM of YB to obtain more information regarding PM bacterial communities.

Hence, this study aimed to investigate the bacterial community structure and diversity in PM of YB using high-throughput sequencing. Furthermore, the physicochemical characteristics of PM were determined to investigate the correlation between bacterial community structure and environmental factors, which will assist us in accumulating basic data for further elucidating the mechanism of Baijiu brewing, and will promote the long-term development of YB.

## MATERIALS & METHODS

### Sampling

PM samples were collected in December 2017 from 13 distilleries located in Yibin (28°76′N, 104°63′E). In total, 68 samples were collected, among which 37 were collected from the middle layer of cellar walls and 31 were collected from the substrate layer (100 g PM at each position). The PM sampling sites are shown in Fig. S1. PM samples were numbered according to the information regarding the distillery, cellar, and layer, and were divided into two groups, the middle-layer and the sub-layer (Table S1). The samples were aseptically collected in sterilized bags and maintained at low temperature during transport from the distilleries to the laboratory.

### Physicochemical properties

The basic physicochemical properties of PM, including moisture content, pH, and available phosphorous, ammonia nitrogen, and humic acid content were also determined as described previously (*Shen, 2014*).

### DNA extraction, polymerase chain reaction (PCR), and Illumina HiSeq sequencing

DNA was extracted from the PM using the FastDNA SPIN kit for soil (Mpbio, USA) according to the manufacturer's instructions. The crude DNA was quantified based on the absorbance at 260 nm using a NanoDrop$^{TM}$ one spectrophotometer (Thermo Fisher Scientific, USA), and the purity was assessed using agarose gel electrophoresis (Beijing Liuyi Biological Technology Co., Ltd. China).

To amplify the V3–V4 regions of 16S rRNA genes, a pair of universal bacterial primers was used in PCR: a forward primer (341F, CCTAYGGGRBGCASCAG) and a reverse primer (806R, GGACTACHVGGGTWTCTAAT). The reaction mixtures (25 μl) contained 5 μl of 5× TransStart Fastpfu buffer (Transgen Biotech), 50 ng DNA sample, 1 μl of each primer, 2 μl 2.5 mM dNTPs, and 0.5 μl TransStart Fastpfu DNA polymerase (Transgen Biotech). The PCR conditions were as follows: 2 min at 95 °C; 30 cycles of 20 s at 95 °C, 30 s at 55 °C, 30 s at 72 °C; final extension at 72 °C for 5 min and cooling at 4 °C. The PCR products were purified using a universal PCR purification kit (Tiangen, Beijin, China) and sent to a commercial sequencing company for high-throughput sequencing using the Illumina Hiseq sequencing platform (MeiYin Health Technology (Beijing) Co., Ltd., China).

### Nucleotide sequence accession numbers

The HiSeq sequencing data were submitted to the Sequence Read Archive (SRA) of the NCBI database as BioProject ID PRJNA597727.

### Data analysis

All the raw HiSeq-generated 16S rRNA gene sequencing data were processed using the QIIME pipeline (V1.9.1). After initial quality control (QC) processing, pairs of reads were merged using FLASH (V1.2.7, *Magoč & Salzberg, 2011*) to acquire raw tags. Then, they were screened using the QIIME function (*Caporaso et al., 2010*) to obtain clean tags. Chimeras were detected and removed using the Uchime algorithm to obtain effective tags (*Edgar et al., 2011*). The effective tags, at a similarity threshold of 97%, were grouped into the same operational taxonomic units (OTUs). A representative sequence for each OTU was extracted and annotated using the SILVA database V119 (*Quast et al., 2012*).

Community richness and diversity were estimated using the Chao1 (*Chao & Bunge, 2002*) and Shannon indices (*Shannon, 1997*), respectively. Bacterial community differences in two groups of PM were evaluated using principal coordinates analysis (PCoA) in Fast UniFrac (*Hamady, Lozupone & Knight, 2010*). Linear discriminant analysis effect size (LEfSe) was used to compare the significance of species-specific differences between the two groups (*Segata et al., 2011*). Canonical correspondence analysis (CCA) between bacterial communities in the PM and physicochemical properties was performed using the vegan package in R (*Oksanen et al., 2018*). All the statistical analyzes, including Spearman's test, were performed with the SPSS software (V22.0, IBM Corp., Armonk, NY, USA). All tests for significance were two-sided, and $P$- values <0.05 were considered statistically significant.

## RESULTS

### Physicochemical properties of the PM

The physicochemical characteristics of the PM of the two groups are shown in Table S2 . Among the physicochemical indicators analyzed using one-way analysis of variance (ANOVA) of SPSS Statistics 22, pH differed significantly ($P < 0.05$) between the two groups. However, no significant differences in moisture content, and ammonia nitrogen, available phosphorus, and humic acid levels were observed.

### Diversity of bacterial community in PM

In total, 68 samples were analyzed for amplicon sequencing. On an average, 99,639 clean tags were generated for each sample with a minimum of 53,888 tags. In total, 53,493–208,410 effective tags for each sample with an average sequence length of 444 bp were generated. Rarefaction curves were used to assess the degree of completion of a taxonomic survey (*Good, 1953*). The rarefaction curve showed that all curves approached the saturation plateau, proving that the identified sequences of PM samples represented almost all the bacterial sequences (Fig. S1).

Eventually, 220,790 operational taxonomic units (OTUs) were detected based on 97% similarity in 16S rRNA gene sequences, of which the middle-layer group and sub-layer group had 186,289 and 148,904 OTUs, respectively. Furthermore, 114,403 OTUs were
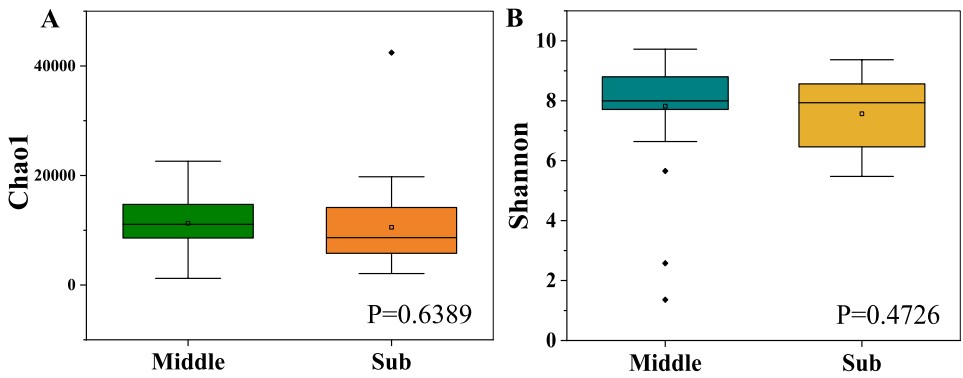

**Figure 1** **Chao1 richness indices (A) and Shannon diversity indices (B).** The box-plots exhibit the first (25%) and third (75%) quartiles, the median, and the maximum and minimum observed values within each data set.

shared between the two groups. The mutual and unique OTUs for the two groups are shown in Venn diagrams (Fig. S2).

The bacterial richness (Chao1) indices for all samples ranged from 1,214.30 to 42,441.82, and were fairly similar in the two different groups, with 11,296.93 ± 4,816.73 in the middle-layer group and 10,549.07 ± 7,638.48 in the sub-layer group. Furthermore, the diversity (Shannon) indices were 7.82 ± 1.68 in the middle-layer group and 7.57 ± 1.21 in the sub-layer group. The Chao1 and Shannon indexes did not vary significantly between the two groups ($P > 0.05$), suggesting similar overall species diversity (Fig. 1). Furthermore, the PCoA analysis showed that there was no explicit clustering of samples within the groups, demonstrating significant inter-individual variations in the bacterial communities (Fig. 2).

## Bacterial community structure in PM

In total, 49 phyla, 332 families, and 811 genera were identified in all the samples (Table 1). The most abundant phylum was Firmicutes (71.9%), followed by Euryarchaeota (11.3%), Bacteroidetes (6.5%), Synergistetes (3.0%), Actinobacteria (1.4%), Proteobacteria (1.2%), Cloacimonetes (0.6%), Tenericutes (0.4%), Chloroflexi (0.3%), and Cyanobacteria (0.2%) (Fig. 3). Most notably, Firmicutes constituted more than 70.0% of the bacterial population in all the groups; in addition, Euryarchaeota and Bacteroidetes were also predominant in all groups with average relative abundances >5%. The proportions of these three major phyla in the middle-layer and sub-layer group were 71.2% and 72.3%, 12.7% and 10.0%, and 6.9% and 6.3%, respectively.

At the genus level, dominate genera were similar between the two groups. Fourteen genera dominated all the samples (average relative abundance >1% in all groups), including *Ruminiclostridium* 5 (17.9%), *Gelria* (8.4%), *Methanoculleus* (4.9%), *Syntrophomonas* (3.8%), *Lactobacillus* (3.6%), *Syntrophaceticus* (2.8%), *Proteiniphilum* (2.7%), *Aminobacterium* (2.7%), *Petrimonas* (2.4%), *Sedimentibacter* (2.1%), *Methanosarcina* (1.9%), *Methanobacterium* (1.8%), *Methanobrevibacter* (1.8%), and *Caldicoprobacter*
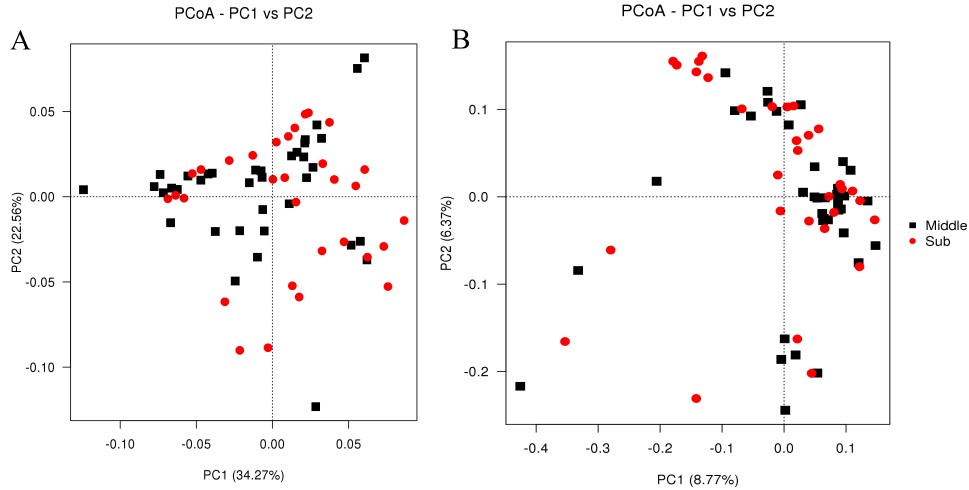

**Figure 2** **PCoA of weighted Unifrac distance (A) and unweighted Unifrac distance (B) for the microbiota at PM of middle-layer group and sub-layer group.** The principal coordinate combination with the highest contribution rate was selected for display.

**Table 1** **Number of classified species at different taxa.**

| Group | Phylum | Class | Order | Family | Genus |
|---|---|---|---|---|---|
| Middle | 45 | 88 | 151 | 291 | 680 |
| Sub | 46 | 88 | 160 | 304 | 713 |
| All | 49 | 94 | 172 | 332 | 811 |

(1.7%). The most abundant 10 genera are shown in Fig. 4. In addition, the relative abundance of *Clostridium sensu stricto 14* and *Clostridium sensu stricto 12* also exceeded 1% in the sub-layer group, being 1.8% and 1.7%, respectively. LEfSe was used to determine the classified bacterial taxa with significant abundance differences between two groups. The results of the LEfSe analysis showed that *Lactobacillus* is the key discrepant bacteria, and that the relative abundance of *Lactobacillus* in the sub-layer PM was significantly higher than that in middle-layer PM (Fig. 5).

## Comparison of dominant bacterial communities in different Baijiu PM

Based on the results of previous studies, we compared the dominant microbiota of strong-aroma Baijiu in different regions to ascertain the common dominant bacteria, and the unique dominant bacteria of YB (Table 2). Strong-aroma Baijiu PM contained four mutually dominant bacteria in five different regions (Mianzhu, Luzhou, Jiangsu, Hunan, Yibin), namely, *Lactobacillus, Sedimentibacter, Syntrophomonas*, and *Methanobrevibacter*. In addition, it is noteworthy that *Petrimonas, Clostridium, Methanosarcina, Methanoculleus*, and *Methanobacterium* were also detected in our study with the relative abundance above 1%, and almost all of them belonged to phylum Euryarchaeota. Three unique bacteria were also detected in YB PM, namely *Ruminiclostridium 5, Gelria*, and *Syntrophaceticus*.

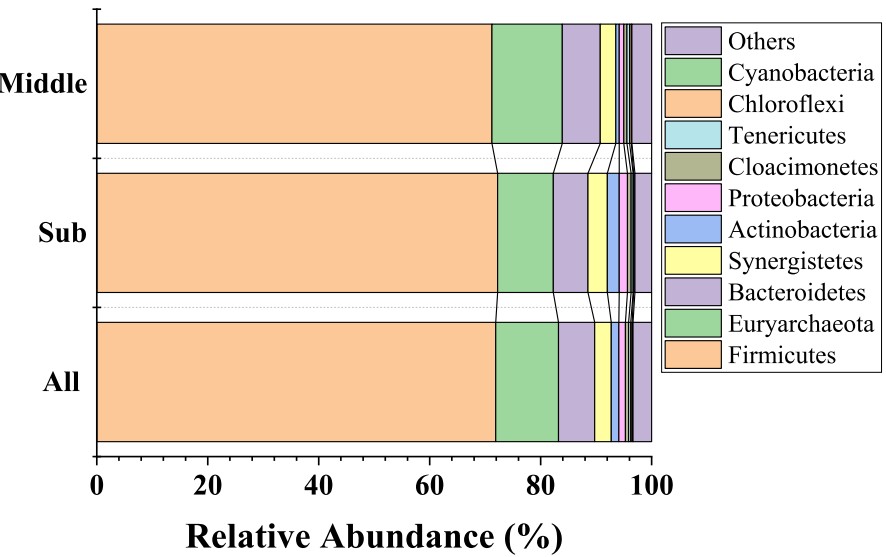

**Figure 3** **Taxonomic classifications of the bacterial communities in PM samples collected from sub-layer and middle-layer at the phylum level.** The "Others" refer to the group that could not be accurately assigned to any known bacterial taxonomic group at the phylum level, as well as the group that ranked after 10 phyla with most relative abundance.

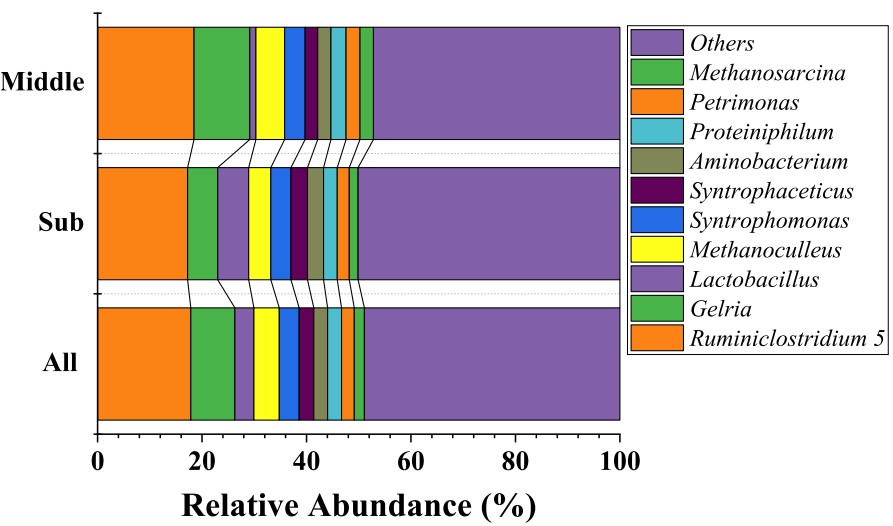

**Figure 4** **The ten most abundant genera in bacterial community at the genus level.**

In addition, the common dominant microorganisms of strong-aroma Baijiu and those of other aroma types (miscellaneous-aroma and sauce-aroma) were compared. The dominant microbiota composition in the Baijiu PM of different aroma types differed considerably. Only *Lactobacillus* was the dominant microorganism shared by three aroma types of Baijiu PM. In addition, *Enterobacter* was the other dominant microorganism shared by miscellaneous-aroma and sauce-aroma, which was not found in strong-aroma Baijiu PM.

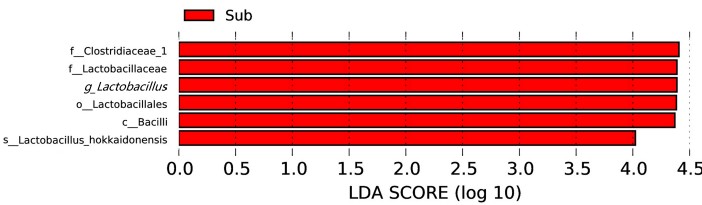

**Figure 5  LDA score map with a threshold value of 4.0.** The length of the histogram represents the effect size of the species with significant difference in abundance in different groups.

**Table 2  Comparison of dominant microorganism in PM in different regions of strong-aroma Baijiu.**

| Locations | Dominant microorganism | Ratio[a] | Reference |
|---|---|---|---|
| Mianzhu (31°43′N, 104°12′E) 300 km | *Lactobacillus, Petrimonas, Clostridium IV, Sedimentibacter, Syntrophomonas, Spirochaetes SHA-4, unclassified Porphyromonadaceae, Anaerobrancaceae, Clostridiaceae 1, Ruminococcaceae, Methanoculleus, Methanosarcina, Methanobacterium, Methanobrevibacter* | 8/14 | *Tao et al. (2014a)* |
| Luzhou (28°53′N, 105° 27′E) 70 km | *Methanobrevibacter, Caproiciproducens, Petrimonas, Lactobacillus, Sedimentibacter, Proteiniphilum, Syntrophomonas, Aminobacterium, Christensenellaceae R-7, Caldicoprobacter, Olsenella* | 8/11 | *Liu et al. (2017a)* |
| Jiangsu (35°20′N 121°57′E) ′ 1.700 km | *Lactobacillus, Ruminococcus, Caloramator, Clostridium, Sedimentibacter, Syntrophomonas, Sporanaerobacter, Pelotomaculum, T78, Prevotella, Blvii28, Methanobacterium, Methanobrevibacter, Methanosaeta, Methanoculleus, Methanosarcina, Nitrososphaera* | 7/17 | *Hu et al. (2016)* |
| Hunan (28°21′N, 109°45′E) 470 km | *Petrimonas, Lactobacillus, Sedimentibacter, Clostridium, Ruminococcus, Syntrophomonas, Symbiobacterium, Methanobacterium, Methanobrevibacter, Methanoculleus Methanosarcina* | 8/11 | *Wang, Du & Xu (2017)* |
| Yibin (28°76′N, 104°63′E) 0 km | *Ruminiclostridium 5, Lactobacillus, Gelria, Methanoculleus, Syntrophomonas, Syntrophaceticus, Aminobacterium, Proteiniphilum, Petrimonas, Sedimentibacter, Methanobrevibacter, Caldicoprobacter, Methanosarcina and Methanobacterium* | 14/14 | This work |
| **Unique** | *Ruminiclostridium 5, Gelria, Syntrophaceticus* | – | This work |
| **Common** | *Lactobacillus, Sedimentibacter, Syntrophomonas, Methanobrevibacter* | – | All |

**Notes.**
[a]Ratio: The ratio of the number of dominant genera which shared with ''Yibin Baijiu'' pit mud to the total number of dominant genera.

The detailed information regarding the dominant microflora of the three aroma types of Baijiu PM is shown in Table 3.

## Correlation between environmental factors and bacterial community

CCA was performed to identify the major environmental factors that affected the variation in bacterial communities. The contribution of environmental factors was as follows: available phosphorous ($R^2 = 0.237$, $P = 0.0005$) > pH ($R^2 = 0.210$, $P = 0.0015$) > humic acid ($R^2 = 0.056$, $P = 0.1559$) > ammonia nitrogen ($R^2 = 0.053$, $P = 0.1724$) > moisture

**Table 3  Comparison of dominant microorganism in PM of different aroma types of Baijiu.**

| Representative aroma | Dominant microorganism | Reference |
|---|---|---|
| miscellaneous-aroma | *Corynebacterium, Myroides, Sphingobacterium, Lactobacillus, Clostridium, Acetobacter, Alcaligenes, Enterobacter and Acinetobacter* | *Huang et al. (2017)* |
| | Clostridiaceae, Ruminococcaceae and Thermoanaerobacteriaceae | *Wang et al. (2015)* |
| sauce-aroma | *Methanoculleus, Methanosarcina, Methanosaeta and Methanobacterium* | *Bian et al. (2012)* |
| | *Weissella, Lactobacillus, Bacillus, Enterobacter and Paenibacillus* | *Wang, Zhang & Liu (2016)* |
| strong-aroma (common) | *Lactobacillus, Sedimentibacter, Syntrophomonas and Methanobrevibacter* | – |

content ($R^2 = 0.049$, $P = 0.1959$). The results of CCA indicated that among these five factors, available phosphorous and pH maximally affected PM bacterial community structure. Furthermore, the Spearman's correlation coefficient was determined between environmental factors and the most abundant 35 genera were performed. The results showed that pH correlated negatively with the relative abundance of *Clostridium sensu stricto 12* ($R = -0.502$, $P = 0.002$), *Lachnoclostridium* ($R = -0.473$, $P = 0.008$), and *Lactobacillus* ($R = -0.451$, $P = 0.020$) at the genus level (Fig. 6).

# DISCUSSION

This is the first study to use the Illumina Hiseq sequencing platform for investigating the population diversity of bacteria in the middle-layer and sub-layer PM of YB. In addition, the relationship between bacterial community structure and environmental factors was also assessed. The results indicated that the dominant bacteria in PM of YB were different from that of other regions and aroma types of Baijiu. The pH and available phosphorus content significantly affected the PM bacterial community structure.

According to previous studies, the total number of effective tags of the 16S rRNA gene of microorganisms in PM obtained from the Roche 454 or Illumina Miseq sequencing platforms were less than 10,000 (*Liu et al., 2017a*; *Tao et al., 2014b*). In our study, more than 200,000 effective tags were detected in a single sample with an average of 94,023 tags for all samples. Thus, the result of sequencing can adequately reflect most of the sample diversity noted.

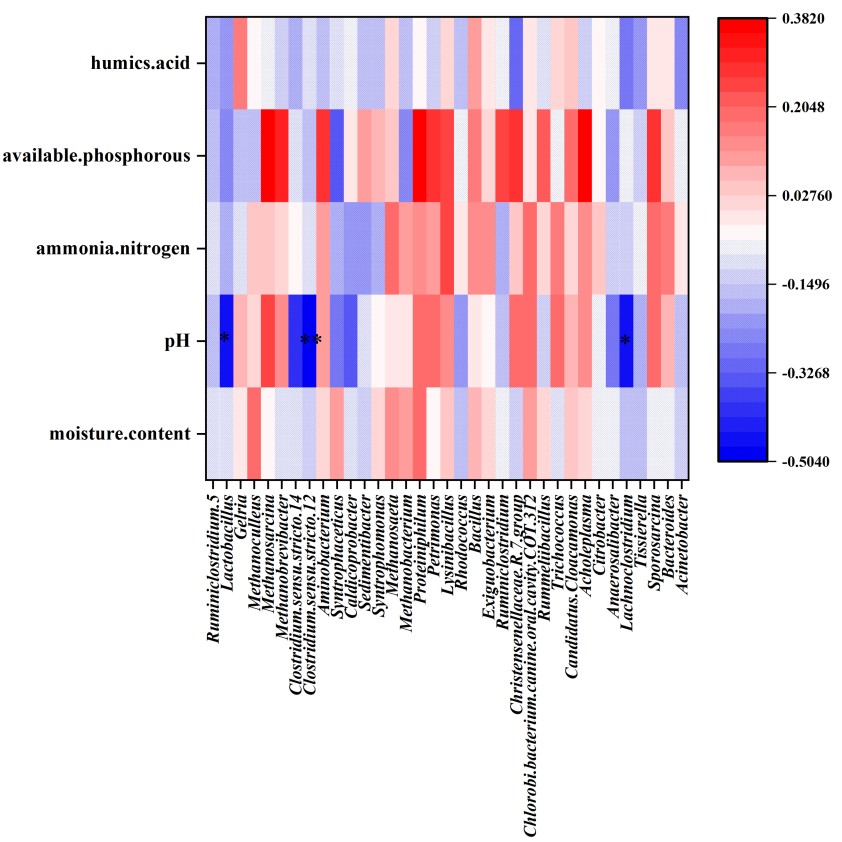

**Figure 6** **Heat map showing the Spearman'.s correlation coefficient of environmental factors and bacterial community in PM.** The horizontal direction shows the species information, the longitudinal direction shows the environmental factor information, the heat map corresponds to the Spearman's correlation coefficient *R*, and the *R* value ranges from −1 to 1. $R < 0$ indicated negative correlation and $R > 0$ indicated positive correlation.

In total, 49 phyla, 332 families and 811 genera were detected, which may be helpful for further understanding the phylogenetic status of numerous uncultured microbes in the PM ecosystem. Although the relative abundance of Firmicutes, Euryarchaeota, Bacteroidetes, and Synergistetes differed slightly between middle-layer and sub-layer samples, they were still the key components of the microbiota in the PM of YB. Previous studies also showed that Firmicutes, Euryarchaeota and Bacteroidetes were the three most abundant phyla in PM (*Liang et al., 2015*; *Liu et al., 2017a*; *Tao et al., 2014a*). Firmicutes include syntropyhic bacteria that can degrade volatile fatty acids such as butyrate and its analogs, and is represented mainly by the *Clostridia* and *Bacilli* (*Garcia-Peña et al., 2011*). Euryarchaeota includes diverse groups of methanogens that are interspersed with non-methanogenic lineages. Methanogens are often symbiotic with hexanoic acid bacteria and can effectively increase the content of ethyl hexanoate in strong-aroma Baijiu via "hydrogen transfer" effect between species (*Zhao et al., 2017*). Bacteroidetes includes three major types of bacteria: Bacteroidia, Flavobacteriia, and Sphingobacteriia. Most of them are capable of

degrading macromolecular carbohydrates such as cellulose to produce acid (*Guo et al., 2014*). All these phyla in PM played a vital role in Baijiu fermentation.

At the genus level, the dominant bacteria in PM from different regions producing strong-aroma Baijiu were compared. In total, 11 dominant genera were detected by *Liu et al. (2017a)* in CSAB's PM collected from Luzhou, Sichuan province (about 110 kilometers from Yibin city), among which 8 genera were also the dominant bacteria in YB PM. *Tao et al. (2014a)* reported 14 dominant genera in CSAB's PM collected from Mianzhu, Sichuan province (about 300 kilometers from Yibin city), 6 of which were different from those of YB PM. In comparison, we observed that the composition of the dominant genera in PM in the Mianzhu area differed from that in the Yibin area due to their geographical distance. In contrast 7 of the 17 dominant genera in Jiangsu area (about 1,700 kilometers away from Yibin city) Baijiu PM were shared by YB PM. The dominant genera in Jiangsu area Baijiu PM and YB PM differed significantly, which further suggested that geographical location considerably affects Baijiu PM bacterial community. Previous studies have demonstrated that the composition of PM bacterial community varied by geographical location (*Huang et al., 2015*; *Liang et al., 2016*). Interestingly, the dominant genera in the Baijiu PM of Hunan region, which is 470 km away from Yibin City, was also similar to the dominant genera in YB PM. It was speculated that the identical latitude of Hunan and Yibin, which resulted in both having the same climate, especially temperature, significantly affected the diversity and structure of the microbiota. However, the farther are two places, the more is the similarity in the dominant genera of the Baijiu PMs from these areas.

The dominant genera in the PM of strong aroma Baijiu differed considerably from those of Baijius of other aromas (Table 3). Obviously, the differences in these dominant bacteria contributed significantly to the formation of different aroma types of Baijiu. *Lactobacillus* was a dominant genus among Baijiu PMs of several aroma types, (*Huang et al., 2017*; *Wang, Zhang & Liu, 2016*; *Hu et al., 2016*), leading to the speculation that *Lactobacillus* was critical for Baijiu brewing. *Gelria* was found to be the dominant genus in our study, which has been rarely reported in previous studies. In syntrophic association with a hydrogenotrophic methanogen, *Gelria* can utilize various amino acids and sugars to produce acetate, propionates, $H_2$, $NH_4^+$, and $CO_2$ (*Plugge et al., 2002*). *Fan et al. (2019)* reported that *Gelria* showed a significant positive correlation with the content of major flavor substances such as alcohols, esters, and aldehydes in Baijiu. Therefore, as a unique dominant genus of YB PM, we speculated that *Gelria* may contribute to the formation of unique flavor of YB. Furthermore, a positive correlation between humic acid content and *Gelria* abundance was evident in the heat map (Fig. 6). Humic acid is an important organic matter in PM, which is formed by the accumulation of microbial metabolites and can completely reflect the ability of PM to supply nutrients to microorganisms (*Zhu et al., 2018*). We speculated that the metabolic activity of *Gelria* promoted the formation of humic acid, which in turn provided nutrition for the growth and metabolism of *Gelria*.

The difference between the physicochemical indicators of the middle-layer and sub-layer PM of YB was analyzed using ANOVA. Results showed that the pH of the sub-layer was significantly lower than that of the middle-layer. Meanwhile, CCA analysis showed that the pH of PM correlated considerably with the bacterial community structure in PM.

Spearman's analysis revealed that the *Clostridium sensu stricto* 12, *Lachnoclostridium,* and *Lactobacillus* were correlated negatively with pH value. LEfSe also showed that *Lactobacillus* and *Clostridiaceae_* 1 were significantly less abundant in the middle-layer of PM with higher pH. In particular, the above three genera that were significantly related to PM pH belonged to Firmicutes, the most abundant phylum. Among them, *Clostridium sensu stricto* 12 and *Lachnoclostridium* belonged to the order Clostridiales, while *Lactobacillus* belonged to the order Lactobacillales. Bacteria of order Clostridiales possess the metabolic ability to convert organic substances into organic acids (such as butyric, caproic acids, etc.), alcohols, $CO_2$ /$H_2$, and minerals (*Kenealy, Cao & Weimer, 1995*; *Zheng et al., 2013*). Furthermore, Clostridiales contributed to the formation of ethyl butyrate and ethyl caproate when butyric and caproic acids reacted with alcohols via enzymatic and non-enzymatic catalysis (*Ding et al., 2014b*). Lactobacillales also plays an important role in Baijiu brewing, as the lactic acid produced can form ethyl lactate. Ethyl lactate, ethyl butyrate, and ethyl caproate are vital flavor compounds in CSAB (*Fan & Qian, 2005*).

Interestingly, a strong correlation of the bacterial community with the available phosphorous level was also observed using CCA analysis. The available phosphorus provides nutrients for the growth and reproduction of microorganisms. Phosphorus is present in the cell nuclear membrane and is a source of the energy substance ATP. *Zheng et al. (2013)* reported that the low pH and high content of available phosphorus in PM may have resulted in the orthogenesis of microorganisms and Clostridiales-, Lactobacillaceae-, and Bacillales-dominated microbial community structure. However, Spearman's analysis showed the absence of any significant correlation between bacterial presence and the available phosphorous level. This may be because Spearman's analysis only performed a pairwise correlation between environmental factors and the most abundant 35 genera and did not reflect the overall situation. In addition, the available phosphorous level may correlate significantly with some low-abundance genera in the PM, which was not evident in the Spearman's heat map.

In this study, we analyzed the community characteristics of bacteria in PM of YB, and observed that fungi also played an indispensable role in fermentation. We will study the community structure and diversity of fungi in the future. Furthermore, using a combination of metagenomic technology and metabolite detection technology, we will investigate the functional microorganisms that affect the quality and flavor of YB in PM. This will provide theoretical guidance and technical support for the application of functional bacteria in YB development.

## CONCLUSIONS

This study demonstrated the population diversity of bacteria in the PM of YB from 13 distilleries using HiSeq sequencing. Our results suggested the presence of a complex community structure and abundant species in the PM of YB. No significant difference was observed in bacterial diversity between middle-layer and sub-layer PM. The LEfSe analysis showed that the relative abundance of *Lactobacillus* in the sub-layer PM was significantly higher than that in middle-layer PM. pH differed significantly ($P < 0.05$) between the two

groups. Available phosphorous and pH strongly affected bacterial community structure in PM. Our study provided basic information for further elucidating the diversity of microbiota in the PM of YB and the potential mechanism of Baijiu production.

### Funding

This study was supported by Yibin Science and Technology Bureau (NO. 2017CNY001-4). The funders had no role in study design, data collection and analysis, decision to publish, or preparation of the manuscript.

### Grant Disclosures

The following grant information was disclosed by the authors:
Yibin Science and Technology Bureau: 2017CNY001-4.

### Competing Interests

The authors declare there are no competing interests.

### Author Contributions

- Li Chen and Yuzhu Li conceived and designed the experiments, performed the experiments, analyzed the data, prepared figures and/or tables, authored or reviewed drafts of the paper, and approved the final draft.
- Lei Jin conceived and designed the experiments, performed the experiments, analyzed the data, prepared figures and/or tables, and approved the final draft.
- Li He, Xiaolin Ao, Shuliang Liu, Yong Yang and Aiping Liu analyzed the data, authored or reviewed drafts of the paper, and approved the final draft.
- Shujuan Chen conceived and designed the experiments, prepared figures and/or tables, and approved the final draft.
- Likou Zou conceived and designed the experiments, authored or reviewed drafts of the paper, and approved the final draft.

### Data Availability

The HiSeq sequencing data are available at the Sequence Read Archive (SRA) of NCBI: PRJNA597727.

### Supplemental Information

Supplemental information for this article can be found online at http://dx.doi.org/10.7717/peerj.9122#supplemental-information.

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
