# Peer review of "Analyzing bacterial community in pit mud of Yibin Baijiu in China using high throughput sequencing"

_PeerJ, doi:10.7717/peerj.9122_

## Round 0.1 · original submission · Major Revisions

Please address carefully all the comments of the reviewers, especially, the first two reviewers that suggested language checking and major revisions for the manuscript in order to be considered for publication.

Reviewer 1 ·

Basic reporting

no comment

Experimental design

no comment

Validity of the findings

no comment

Additional comments

Chen et al. reported the bacterial community in pit mud of Yibin Baijiu by high throughput sequencing. They analyzed 68 samples from 13 baijiu enterprises with two kinds of samples types consisting of middle layer and substrate layer. And the results are easily to predict, and the author should dig deeply based on the data from HTS.
The following questions should be concerned.
1. Line 25 and the whole manuscript, the 16S rRNA high throughput sequencing is not a correct description, and it should be 16S rDNA high throughput sequencing. The author should correct them and the associated description carefully.
2. The author should pay more attention at manuscript writing, such as: Line 97 the “sterilized bas” should be “sterilized bags”; Line 159: “indices was” should be “were”.
3. Can the author explain what are middle layer and substrate layer (the author can draw a picture to make it easier to know), and why the sample numbers are not the same between middle layer (37) and substrate layer (31)?
4. I think the Table 1 should not be shown in manuscript, and it can be replaced by Table S1. The sd values of Table 1 are too big, which means that analyzing the 13 baijiu enterprises in a whole is not correct, and it should be listed as Table S1.
5. Line 253, “Among baijiu PM of several aroma types, Lactobacillus was a dominant genus”, the author should give a reference.
6. Line 260-261. “Therefore, as a unique dominant genus of “Yibin Baijiu” PM, we speculated that Gelria may contribute to the formation of unique flavour of “Yibin Baijiu”.” The author speculated that Gelria may contribute to the formation of unique flavour of Yibin Baijiu, and I have a different opinion that Gelria may contribute to the formation of humics acid according to Fig. 6.
7. I even do not know what is the aim of this manuscript.
a. According to the Introduction from line 79 to 89, it seems that the author wants to investigate many enterprises that produce CSAB, and find something interesting. However, the author only investigated enterprises in Yibin city, and I think the author should investigate enterprises that produce CSAB in China. Because the 13 baijiu enterprises in Yibin city are too close in distance.
b. At the same time, the author sampled in middle layer and substrate layer of the PM, and the difference between middle layer and substrate layer was not analyzed with the variation of baijiu enterprise. The author compared the two groups in a whole not in every baijiu enterprise. Therefore, there is no difference between the two groups. I think the author should deeply analyze the data they got from HTS.
c. The author focused on the PM of Chines strong-aroma baijiu (CSAB) in introduction. however, the author discussed a lot of other aroma types PM at line 239-251.

Reviewer 2 ·

Basic reporting

The presented article concerns the analysis of a popular liqueur from China. This is a traditional drink that has already been analyzed in many ways.
The text is interesting but requires thorough corrections.
An introduction does not explain the purpose of the study sufficiently and should be written in more academic style. The text should go through language correction. Examples: L97: sterilized bas (?), L148: Rarefaction curves was (?), "origined" or "orginated" etc.

Experimental design

Methods are used correctly, but I have some comments:
1. Physicochemical properties were performed according to AOAC ?
2. In methodology are added websites links, pleas edit it in a different way
3. Which statistical test was used? In discussion is a piece of information about "t-test" is this "Student's t-test"?

Validity of the findings

The topic is new, but the results obtained describe a liqueur from a specific region. There is no justification in the text of a stronger need for such research. Conclusions are clear and well described.

Additional comments

The text should be language-checked. The methodology should be revised to incorporate the comments. The need for stronger justification for the research.

Reviewer 3 ·

Basic reporting

The whole manuscript was basically clear and English is generally good. But the authors should give more description on the defined comprise of fermentation pits in 13 baijiu enterprises, Is there something substentially different in thefermentation pits for the four types of aroma types?

Experimental design

The research questions is clear.

Validity of the findings

The results are clearly stated.

Additional comments

I think the manuscript submitted by Chen et al. descrbied the bacterial community in pit mud of Yibin Baijiu in China by high throughput sequencing. the bacterial community from 68 samples 13 baijiu enterprises in Yibin was analyzed and its correlation with environmental factors.
1. In the introduction section, authors should give more description on the defined comprise of fermentation pits in 13 baijiu enterprises.
2. In table 3, why the author bring the strong-aroma out.
Some minor modifications need to be made:
16S not 16s; Table 1 mg/100g, please leave a space between the number and unit. P for the significance should be italic.

---

## Round 0.2 · accepted · Accept

Many grammatical or typographical errors have been revised and the paper has been approved by a native speaking professional Editor. All the reviewers' comments have been addressed correctly.